# 25-Hydroxycholesterol 3-Sulfate Recovers Acetaminophen Induced Acute Liver Injury via Stabilizing Mitochondria in Mouse Models

**DOI:** 10.3390/cells10113027

**Published:** 2021-11-05

**Authors:** Yaping Wang, William M. Pandak, Edward J. Lesnefsky, Phillip B. Hylemon, Shunlin Ren

**Affiliations:** Department of Internal Medicine, McGuire Veterans Affairs Medical Center, Virginia Commonwealth University, Richmond, VA 23249, USA; yaping.wang@vcuhealth.org (Y.W.); william.pandak@va.gov (W.M.P.); edward.lesnefsky@vcuhealth.org (E.J.L.); Phillip.Hylemon@vcuhealth.org (P.B.H.)

**Keywords:** 25HC3S, oxysterol sulfation, acute liver injury, acetaminophen, DNA CpG methylation

## Abstract

Acetaminophen (APAP) overdose is one of the most frequent causes of acute liver failure (ALF). N-acetylcysteine (NAC) is currently being used as part of the standard care in the clinic but its usage has been limited in severe cases, in which liver transplantation becomes the only treatment option. Therefore, there still is a need for a specific and effective therapy for APAP induced ALF. In the current study, we have demonstrated that treatment with 25-Hydroxycholesterol 3-Sulfate (25HC3S) not only significantly reduced mortality but also decreased the plasma levels of liver injury markers, including LDH, AST, and ALT, in APAP overdosed mouse models. 25HC3S also decreased the expression of those genes involved in cell apoptosis, stabilized mitochondrial polarization, and significantly decreased the levels of oxidants, malondialdehyde (MDA), and reactive oxygen species (ROS). Whole genome bisulfite sequencing analysis showed that 25HC3S increased demethylation of ^5m^CpG in key promoter regions and thereby increased the expression of those genes involved in MAPK-ERK and PI3K-Akt signaling pathways. We concluded that 25HC3S may alleviate APAP induced liver injury via up-regulating the master signaling pathways and maintaining mitochondrial membrane polarization. The results suggest that 25HC3S treatment facilitates the recovery and significantly decreases the mortality of APAP induced acute liver injury and has a synergistic effect with NAC in propylene glycol (PG) for the injury.

## 1. Introduction

Acute liver failure (ALF) involves the rapid loss of liver function [1]. The clinical presentation of ALF usually includes liver dysfunction, coagulopathy, development of encephalopathy, multi organ failure, and death in over 50% of the cases [2]. Histologically, patients with ALF develop hepatic inflammation leading to fulminant hepatic necrosis and apoptosis. Causes of ALF include, but are not limited to, drug toxicity, viruses, toxins, and ischemia [3]. In the United States and Western Europe, over 50% of all cases of ALF have been attributed to drug-induced hepatotoxicity, especially, acetaminophen (APAP, also known as paracetamol) overdose-induced ALF, which exceeds other drugs by a 4:1 ratio. However, the overall rarity of ALF has limited experimental information to guide its supportive care [4]. Although NAC has been successfully used in the clinic in some cases, liver transplantation remains the only treatment option in severe cases. Hence, there is an unmet medical need to develop an effective therapy for ALF.

APAP has widely been used to treat fever and mild to moderate pain [5]. Hepatocellular death in APAP induced ALF is the result of the formation of highly toxic intermediate N-acetyl-*p*-benzoquinoneimine (NAPQI) [6,7]. The primary metabolic pathway for APAP is glucuronidation and sulfation, which yields relatively non-toxic metabolites that are excreted via the biliary system [8,9]. However, a small amount of the drug can be metabolized via cytochrome P-450 and yield NAPQI, which can be inactivated by conjugating to glutathione under normal circumstances [10]. When APAP is overdosed at toxic levels (generally ≥7.5 g–10 g in an average adult), glucuronidation and sulfation metabolic pathways are saturated and more NAPQI is produced, which may result in glutathione depletion. NAPQI results in increased mitochondrial permeability through formation of protein adducts by binding to cysteine groups on mitochondrial proteins and ion channels [11,12]. This mitochondrial stress or depolarization results in dysfunction of ATP production, imbalance of cellular ions, leakage of mitochondrial cytochrome c into the cytosol, and eventually cell apoptosis and necrosis [13,14,15].

Oxysterols are oxidized forms of cholesterol that are important in many biological processes including: cholesterol homeostasis, atherosclerosis, platelet aggregation, and apoptosis [16,17]. 25-hydroxycholesterol (25HC), an oxysterol biosynthesized from cholesterol by CYP27A1, can be sulfated by SULT2B to generate 25-hydroxycholesterol 3-sulfate (25HC3S) [18,19]. 25HC3S has been reported to suppress inflammatory responses, inhibit cellular apoptosis, and improve cellular survival [20,21,22,23,24,25,26,27,28]. As reported previously, administration of 25HC3S significantly alleviated injury in multiple organs and reduced mortality in the lipopolysaccharide (LPS)-induced endotoxin shock mouse model [29]. Recent studies have shown that 25HC and 25HC3S served as paired epigenetic regulators, playing an important role in global gene regulation by methylating and demethylating ^5m^CpG in key promoter regions involved in many cellular signaling pathways [30]. Regulation of gene expression via demethylation of ^5m^CpG in promoter regions may be the primary mechanism by which 25HC3S decreases lipid accumulation, reduces inflammation, and increases cell survival.

In the current study, we explored the effect of 25HC3S in the APAP-induced ALF and organ injury mouse models. The results showed that 25HC3S significantly decreased mortality, improved hepatic function, increased mitochondrial polarization, and reduced the levels of oxidants and cell death (especially apoptosis) following APAP overdose. These activities of 25HC3S appeared to be mediated by demethylation of ^5m^CpG in key promoter regions of genes involved in MAPK-ERK and PI3K-Akt cell signaling pathways.

## 2. Materials and Methods

### 2.1. Materials

APAP was purchased from Sigma-Aldrich (St. Louis, MO, USA). 25-Hydroxycholesterol was commercially sourced from Steraloids Inc. (Newport, RI, USA). 25HC3S was synthesized and purified in our laboratory as previously described [22]. The reagents for real-time RT-PCR were obtained from Applied Biosystems (Applied Biosystems, Foster City, CA, USA). The RT^2^ Profiler PCR Array-Cell Death Pathway Finder was acquired from QIAGEN (Valencia, CA, USA). MitoProbe JC-1 Assay Kit for Flow Cytometry and H2DCFDA were purchased from Life Technologies (Carlsbad, CA, USA). Propylene glycol (PG), hydroxypropyl-β-cyclodextrin (HBC), paraformaldehyde, phosphate buffer, 2-thiobarbituric acid (TBA) and malondialdehyde (MDA) were purchased from Sigma-Aldrich (St. Louis, MO, USA). Cell culture media were obtained from Invitrogen (Carlsbad, CA, USA). All other reagents were from Sigma-Aldrich unless otherwise indicated.

### 2.2. Animal Studies

Animal studies were approved by the Institutional Animal Care and Use Committee of McGuire Veterans Affairs Medical Center and were conducted in accordance with the Declaration of Helsinki, the Guide for the Care and Use of Laboratory Animals, and all applicable regulations. Two mouse models, 350 mg and 600 mg/kg of APAP, were used: (1) To study the effect of 25HC3S on liver injury induced by APAP overdose, 12-week-old male C57BL/6J mice (Jackson Laboratory, Bar Harbor, ME, USA) were weight-pair assigned into three groups, control, vehicle, and 25HC3S groups. All mice were intraperitoneally (IP) injected with 350 mg/kg APAP (dissolved in 10% glucose/water at 14 mg/mL) [31]. At −2 h, −1 h, 0 h, +30 min, +1 h or +2 h before, on, or after challenge with APAP, the control group of mice was intravenously (IV) injected with 10% glucose in sterile water, the vehicle had 20% PG and 4% HBC in 10% glucose/water, and the 25HC3S group had 25 mg/kg of the drug in vehicle. (2) For the mortality experiment, 12-week-old female mice were weight-pair assigned into three groups with each receiving IV injection of control, vehicle, or 25HC3S (25 mg/kg) 2 h before IP injection of 600 mg/kg APAP in sterile 10% glucose water. All mice were housed under identical conditions in an aseptic facility with a 12-h light/12-h dark cycle and given free access to water and food. Blood and tissue samples were collected at 24 h after APAP injection under anesthesia. Serum enzymatic activities of alkaline phosphatase (ALK), alanine aminotransferase (ALT), aspartate aminotransferase (AST), and lactate dehydrogenase (LDH) were measured in the clinical laboratory at McGuire Veterans Affairs Medical Center. Mouse survival was monitored every 2 h during the daytime and 12 h during the night.

### 2.3. Histological Analysis

Three specimens from different regions of the liver/lung/kidney of each mouse were collected and fixed in 10% paraformaldehyde in 0.1 M phosphate buffer at room temperature overnight. The regions of the specimens were standardized for all mice. The paraffin-embedded tissue sections (4 μm) were prepared by the Department of Pathology, School of Medicine, Virginia Commonwealth University, then deparaffinized and stained using a standard hematoxylin and eosin (H&E) method [29].

Ten images per sample were taken at ×400 magnification by light microscope and scored by two pathologists in a blinded manner. The severity of microscopic lung injury was graded from 0 (normal) to 3 (severe) based on the degree or amount of (a) congestion of alveolar septae; (b) alveolar hemorrhage; (c) intra-alveolar fibrin; (d) intra-alveolar infiltrates. The total injury score made up of four components was computed for each mouse. The degree of liver injury was determined by the percentage of hepatic parenchyma with apoptosis/necrosis or inflammation and graded on a sliding scale of: 0, absent; 0.5, minimal; 1, mild; 1.5, mild-to-moderate; 2, moderate; 2.5, moderate-to-marked; and 3, marked [32]. Renal tubular injury was assessed using a score in which the percentage of cortical tubules showing epithelial necrosis was assigned a score of either 0, none; 1, <10%; 2, 10–25%; 3, 25–75%; or 4, >75% [29].

### 2.4. Quantitative Real-Time Polymerase Chain Reaction (qRT-PCR) Analysis

The relative mRNA levels were measured by real-time reverse transcriptase polymerase chain reaction as previously described [29]. Briefly, total RNA was isolated with an SV Total RNA Isolation Kit (Promega, Madison, WI, USA) that included DNase I treatment. About 2 µg total RNA was used in the first-strand cDNA synthesis as recommended by the manufacturer (Invitrogen, Carlsbad, CA, USA). Quantitative RT-PCR was performed using SYBR Green as an indicator in an ABI 7500 Fast Real-Time PCR System (Applied Biosystems, Foster City, CA, USA). The amplification of *Rn18s* was used as an internal control. The sequences of primers were designed as recommended by the https://pga.mgh.harvard.edu/primerbank/ (accessed on 17 August 2020) and are summarized in Appendix A.

### 2.5. RT^2^ Profiler PCR Array (Cell Death Pathway Finder) Analysis

Cell death is the major cause of organ injury during the process of endotoxin shock. The mouse Cell Death Pathway Finder RT^2^ Profiler PCR Array was used in the present study to identify the changes in expression of genes involved in the cell death pathways. This PCR array was to detect the 84 key gene expression (mRNA levels), which are involved in central mechanisms of cellular death including apoptosis, autophagy, and necrosis. Twenty-four hours after overdose with APAP (with or without PG or 25HC3S), liver tissues were collected from the mice, and total RNA was isolated from 0.1 g tissue as described previously [29]. After reverse transcription into first-strand cDNA with the RT^2^ First Strand Kit (QIAGEN, Hilden, Germany), all samples were subjected to the RT^2^ Profiler PCR Array Cell Death Pathway Finder assay following the manufacturer’s instructions. The Qiagen PCR Array Data Analysis complementary web-based software was used for analysis and interpreting array data.

### 2.6. Analysis of Whole Genome Bisulfite Sequencing (WGBS)

Genomic DNAs were extracted from 25 mg liver tissues, which were collected from normal, vehicle, or 25HC3S treated mice using QIAamp DNA Mini Kit (QIAGEN, Hilden, Germany). Each sample, 6 µg, was sent to Novogene Co., Ltd. (Tianjin, China) for analysis of whole genome bisulfite sequencing (WGBS). Each sample, 5.2 g of genomic DNA spiked with 26 ng lambda DNA was fragmented by sonication to 200–300 bp with Covaris S220, followed by end repair and adenylation. Cytosine-methylated barcodes were ligated to sonicated DNA as manufacturer’s instructions. These DNA fragments were treated twice with bisulfate using EZ DNA Methylation-Gold TM Kit (Zymo Research, Irvine, CA, USA) before the resulting single-strand DNA fragments were PCR amplified using KAPA HiFi Hot Start Uracil and Ready Mix (2×). Library concentration was quantified by Qubit^®^ 2.0 Fluorometer (Life Technologies, Carlsbad, CA, USA) and quantitative PCR, and the insert size was assayed using an Agilent Bioanalyzer 2100 system.

The library preparations were sequenced using an Illumina Hiseq 2500/4000 or Novaseq platform and 125 bp/150 bp paired-end reads were generated. Image analysis and base calling were performed with Illumina CASAVA pipeline, and finally 125 bp/150 bp paired-end reads were generated. Trimmomatic (v0.36) software was used for quality control. Bismark software (version 0.16.3; Krueger F, 2011) was used to perform alignments of bisulfite-treated reads to a reference genome (X 700-dovetail). DSS software (DSS 2.34.0) was used to identify differentially methylated regions (DMRs). KOBAS software (KOBAS 2.0)was used to test the statistical enrichment of DMR related genes in the Kyoto Encyclopedia of Genes and Genomes (KEGG) pathways [33].

### 2.7. Analysis of Mitochondrial Potential

Changes in mitochondrial membrane potential (MMP) were measured using the MitoProbe JC-1 assay kit for flow cytometry (Invitrogen, Carlsbad, CA, USA). The Huh-7 cell line was routinely grown in DMEM containing 10% fetal bovine serum (FBS), 100 IU/mL penicillin, and 100 mg/mL streptomycin and incubated at 37 °C in a 0.5% CO_2_ incubator. For the MMP experiment, Huh-7 cells were first seeded at a density of 6.5 × 10^5^ per dish in 60 mm dishes. Twenty-four hours after seeding, cells were pretreated with the indicated concentrations of 25HC3S and/or vehicle for 2 h before 10 mM APAP was added [34]. Twenty-four hours after APAP addition, media were removed, and the cells were trypsinized and resuspended in PBS (Invitrogen, Carlsbad, CA, USA). Mitochondria were stained by JC-1 according to the manufacturer’s instructions, and the fluorescence was detected and measured by fluorescence-activated cell sorting (Virginia Commonwealth University FACS Shared Core).

### 2.8. Measurement of Intracellular ROS

The amount of intracellular reactive oxygen species (ROS) in vitro was measured using H2DCFDA (2′,7′-Dichlorodihydro fluorescein diacetate) as an indicator for ROS in cells [35]. Huh-7 cells were routinely grown in DMEM containing 10% FBS, 100 IU/mL penicillin, and 100 mg/mL streptomycin and incubated at 37 °C in a 0.5% CO_2_ incubator. For the ROS detection, Huh-7 cells were first seeded at a density of 6.5 × 10^5^ per dish in 60 mm dishes. Twenty-four hours after seeding, cells were pretreated with 50 µM 25HC3S and/or vehicle for 2 h before 10 mM APAP was added. Sixteen hours after APAP addition, media were removed, and the cells were trypsinized and resuspended in PBS. H2DCFDA was then added to the suspended cells at a final concentration of 10 µM in the dark in an incubator for 30 min and immediately used for ROS detection by flow cytometry at an excitation/emission wavelength of 485/530 nm. Results were also expressed as the percentage increased relative to untreated cells.

### 2.9. Hepatic Lipid Peroxidation (Malondialdehyde, MDA) Assay

Lipid peroxidation of liver in mice was evaluated by measuring the thiobarbituric acid (TBA) according to the modified method by Ohkawa and Mihara [36,37]. Briefly, liver tissue (~100 mg) was homogenized in 1 mL PBS containing 1 mM EDTA and centrifuged at 500× *g* for 10 min at 4 °C. To each 0.5 mL of 10% homogenate of the tissue sample, add 3 mL of 1% H_3_PO_4_ and 1 mL of 0.6% TBA aqueous solution: stir and heat the mixture on a boiling water bath for 45 min. After cooling, add 4 mL of *n*-butanol, shake, and separate the butanol layer by centrifugation; determine the optical density of the butanol layer at 535 and 520 nm; calculate the difference of optical density between the two determinations to be taken as the TBA value. MDA levels were normalized to the hepatic cell protein content as determined by the bicinchoninic acid assay kit purchased from Pierce (Rockford, IL, USA). The amount of lipid peroxidation was expressed as nmol/mg protein.

### 2.10. Statistical Analysis

Data were reported as the mean ± standard deviation (S.D.) and subjected to one-way ANOVA with posthoc Tukey analysis (for multiple groups comparisons and normal distribution). An F test or the Student–Neuman–Keuls post-hoc test analyses were performed on these data to analyze the variances and significances between groups (for two group comparison, two-sided). The Kaplan–Meier Log-Rank test was used for survival analysis. All analyses were performed with SPSS software version 19 for Macintosh. Statistical significance was defined as *p* < 0.05.

## 3. Results

### 3.1. 25HC3S Alleviates Injured Liver Function and Increases Survival Rates in APAP Mouse Model

In order to determine the effect of 25HC3S on liver injury in APAP challenged mice, 12-week-old male C57BL/6J mice were weight-pair assigned into three groups, the control, the vehicle, and the 25HC3S. To avoid the liver damage caused by starving, 10% glucose was used in APAP solution, which gave more consistent results (data not shown), indicating this is a better model. For the mortality experiment, each group of mice was treated with control (10% glucose), the vehicle (or PG), or 25HC3S (25 mg/kg) by IV injection 2 h before IP injection with 600 mg/kg APAP. A global examination of liver tissues showed that APAP induced tissue injury while 25HC3S minimized it (Figure 1A). In 25HC3S pre-treated mice, the survival rate and survival interval were significantly higher than that of both the control and the PG groups (*p* values were 0.0174 and 0.025, respectively). However, post-treatment showed slight decreases in the rate of mortality but not a significant difference between 25HC3S and other groups (data not shown). Interestingly, the survival rate and survival interval of the PG (vehicle) group were also higher than those in the control group (*p* value was 0.05) although was much lower than the 25HC3S group (Figure 1B). For studies of effects on the liver injury, 3 groups of mice were treated with control (*n* = 14), vehicle, or 25 mg/kg 25HC3S in vehicle −2 h, −1 h, 0 h, 30 min, +1 h and +2 h before, on, and after IP injection of 350 mg/kg APAP. Serum enzymatic activities of ALK, AST, and LDH were measured at 24 h after APAP injection. The earlier treatment, the lower levels of the serum markers are observed (data not shown). For clinical usage, the later treatment after the challenge of APAP will be more significant. The best latest treatment is the administration of 25HC3S at 30 min after APAP as shown in Figure 1C–E. Compared to the control group, both PG and 25HC3S treatment significantly reduced serum levels of ALT, AST and LDH by Kruskal–Wallis statistic test. Compared to the vehicle group, 25HC3S treatment had lower but not statistically significant levels of serum ALT, AST and LDH (*p* values are 0.0706, 0.1239 and 0.1410, respectively). The results showed that both PG and 25HC3S alleviated liver injury or improved hepatic function at the lower dose of APAP challenge, but 25HC3S in PG provided a better outcome and with significantly decreased mortality at the higher dose.

NAC is currently used as part of the standard of care in APAP overdose [38]. The effect of N-acetylcysteine and propylene glycol (NAC+PG) with or without 25HC3S on the recovery of hepatic function following APAP overdose were compared. As shown in Figure 1F, NAC+PG decreased serum levels of LDH, AST, and ALT after APAP overdose with *p* values of 0.04, 0.05, and 0.2, respectively. NAC alone (without PG) also reduced these liver enzymes but not statistically significant in LDH while more so in ALT (*p* values of 0.06, 0.05, and 0.007, respectively). The addition of 25HC3S to NAC+PG virtually restored LDH, AST, and ALT to the normal levels with *p* values of 0.015, 0.01, and 0.002, respectively (Figure 1F), indicating that the combination has potential as an optimal therapy of APAP induced acute liver injury.

To confirm the effect of PG and 25HC3S+PG on the recovery of damaged tissues in APAP overdosed mice, tissues of the liver, lung, and kidney were examined by histopathology. All the tissues were severely damaged following the administration of APAP (600 mg/kg), demonstrated by overt infiltration of neutrophils, marked cellular necrosis, and profound structural destruction (Figure 2A), consistent with published results [39,40]. Compared to the control group, the tissue injury scores in groups pretreated with PG or 25HC3S+PG were significantly reduced. Furthermore, the tissues from the 25HC3S+PG group showed normal-like tissue structures and had much lower tissue injury scores, demonstrating that 25HC3S prevented tissue injury in APAP overdosed mice (Figure 2A,B).

### 3.2. 25HC3S Suppresses Apoptosis-Related Gene Expression in the APAP Induced Liver Injury

To better understand the underlying mechanism behind the effects of PG and 25HC3S on APAP toxicity, an RT^2^ Profiler PCR Array of Mouse Cell Death Pathway was used to study the gene expression profile involved in cell apoptosis, necrosis, and autophagy. The expression of 84 genes involved in cell death in the liver was examined (Figure 3). Based on similarity of gene expression, clustergram analysis showed that the expression patterns between vehicle treated and control mice (APAP only) were similar but significantly different from those of normal mice (Figure 3A, lanes P and C vs. N). Interestingly, the patterns from the liver of 25HC3S treated mice were similar to normal mice but significantly different from those of control mice or vehicle (PG) mice (Figure 3A lanes P+S vs. N). Compared to the control group, scatter plot analysis showed that PG treatment both increased and decreased the expression of only one gene (Figure 3B), however, 25HC3S increased the expression of 4 genes and decreased that of 16 genes (Figure 3C). Compared to the PG group, 25HC3S treatment increased the expression of 2 and decreased 10 genes (>2-fold) (Figure 3D). The detailed results are summarized in Appendix A. The array results were confirmed by qRT-PCR as shown in Figure 3E. The expression of pro-inflammatory cytokine genes was determined by qRT-PCR analysis, as shown in Figure 3F. 25HC3S significantly decreased the expression of NFkB and IL-1, consistent with previous reports [21,24], as well as those genes involved in pro-apoptosis or inflammation. Meanwhile, 25HC3S increased the expression of genes involved in cell survival (anti-apoptosis) or autophagy. These results indicated that 25HC3S prevented APAP-induced cell death through the different pathway(s) or mechanisms from that of PG.

### 3.3. 25HC3S Increases Anti-Apoptosis Gene Expression via DNA ^5m^CpG Demethylation

To understand the possible function of cytosine methylation in 25HC3S treated APAP mice, the genomic DNA from the liver tissues were extracted for the construction of bisulfite-treated genomic DNA libraries. In these two libraries, more than 77% of cytosine residues were covered by at least ten reads in “GRCm38”. The depth and density of the sequencing were enough for a high-quality genome-wide methylation analysis. Meanwhile, the efficiencies of bisulfite conversion, represented by the lambda DNA to the libraries, were over 99%, providing reliable and accurate results for the WGBS (Appendix A).

A total 2911 differential methylated regions (DMRs) under CG context were identified as hypomethylated regions located in 939 genes (differential methylated genes, DMGs), among which 44% (414) of the DMGs were identified in their promoters (Figure 4A) following 25HC3S treatment. The hypomethylated genes were highly enriched in 55 KEGG pathways (*p* < 0.05) (Appendix A). The top 22 pathways (*p* < 0.02) were shown in Figure 4B. While no hypermethylated genes were significantly enriched into any of KEGG pathways. Among these pathways, PI3K-Akt and MAPK signaling pathways are believed to be the master pathways regulating cell proliferation and cell death. The chromosome and sequence location of the hypomethylated CpG by 25HC3S in promoter regions of the key genes involved in PI3K-Akt and MAPK signaling pathway are summarized in Table 1 and Table 2, respectively. The results suggest that the effects of 25HC3S on the APAP induced hepatic injury are most likely mediated by demethylation of ^5m^CpG in promoter regions of the key genes, such as the Fgf11, Pik3cb, Pdgfa, Pdgfb involved in PI3K-Akt and MAPK signaling pathways.

### 3.4. Relationship between ^5m^CpG Demethylation in Promoter Regions and Gene Expression

The relationship between gene expression and CpG methylation/demethylation have been well documented. To examine the gene expression as the function of hypomethylation in the promoter regions, genes *Fgf11*, *Pdgfa*, *Pdgfb*, *Map3k6*, *Map4k4*, *Ywhaz*, *Tgfbr*, *Tlr4*, *Pik3cb*, and *Mapk1* in the MAPK and PI3K-Akt pathways were confirmed by RT-PCR analysis. As expected, 25HC3S treatment increased expression of *Pdgfb* by 6.8-fold, *Map3k6* 1.2-fold, *Map4k4* 1.9-fold, *Tgfbr1* 4.2-fold, *Tlr4* 2.2-fold, *Pik3cb* 2.3-fold, *Fgf11* 1.5-fold, *Ywhaz* 1.3-fold, *Pdgfa* 1.3-fold, and *Mapk1* 2.6-fold. Compared with vehicle group, the expression of genes involved in PI3K-Akt (Figure 4C) and MAPK (Figure 4D) signaling pathways were significantly increased in the 25HC3S treated group (*p* < 0.05).

### 3.5. 25HC3S Stabilize the Mitochondrial Polarization

Mitochondrial depolarization and leakage resulting from the mitochondrial permeability transition (MPT) is a key step for necrosis and apoptosis [41]. The MPT blockers, such as cyclosporine A, prevent the loss of the mitochondrial membrane potential and onset of cell death. It has been hypothesized that MPT is a causative mechanism of acute necrotic cell death [42].

Mitochondrial dysfunction, including loss of mitochondrial membrane potential (MMP), has been proposed as the main sub cellular mechanism in hepatotoxicity induced by APAP overdose [6]. In order to assess the effect of 25HC3S on MMP of hepatocytes, Huh-7 cells were treated as indicated in the Methods section. Consistent with the in vivo data, 25HC3S prevented loss of MMP, as shown by JC-1 assay (Figure 5A). APAP treatment resulted in loss of MMP by 30% (compared to normal cell), both PG and 25HC3S+PG minimized loss of MMP dose-dependently with 25HC3S+PG showing better protective activity.

The production and release of ROS (reactive oxygen species) from damaged mitochondria are critical in oxidative damage in pathogenesis and contribute to retrograde redox signaling from the organelle to the cytosol and nucleus [43,44]. ROS degrades polyunsaturated lipids, forming malondialdehyde (MDA) [45]. MDA is a highly reactive compound that occurs as an enol form [46]. It occurs naturally and is a marker for oxidative stress. To confirm the effect of 25HC3S on mitochondrial polarization, the levels of ROS and MDA in liver tissues were determined. As expected, APAP overdose significantly increased ROS by 40% and MDA by 30-fold. 25HC3S treatment restored the levels of both to the normal as shown in Figure 5B,C with a slight increase of GSH levels (data not shown). The results demonstrate that 25HC3S decreased ROS and MDA by maintaining the polarization of the mitochondrial membrane and integrity of the important cellular organelle.

## 4. Discussion

This study helped to elucidate the mechanisms by which 25HC3S recovered APAP induced acute multiple organ injury, including lung, kidney, and especially the liver, and decreased mortality in mouse models. 25HC3S has been shown to be an endogenous DNMT inhibitor and the key regulator in gene regulation [18]. The data from the present study showed that 25HC3S stabilized mitochondrial polarization, decreased the levels of intracellular oxidants, and promoted recovery of hepatic function via DNA ^5m^CpG demethylation in promoter regions of genes involved in important PI3K-Akt and MAPK signaling pathways. These results, based on large data analysis from both in vitro and a well-studied animal model of APAP overdose, present the mechanism by which 25HC3S regulates critical cell signaling pathways to prevent acute organ injury.

The present results also provide strong evidence that the combination of 25HC3S+NAC in PG solution may have synergistic effects on the recovery of APAP induced acute organ injury as shown in Figure 6A, especially the liver injury. NAC is a potent reducing agent and at high concentrations is able to increase GSH, which neutralizes the toxic metabolite NAPQI. NAPQI causes cell death via depolarization of mitochondria. NAC increases glutathione regeneration and enhances the detoxification metabolic pathway of APAP [47]. NAC is the current standard of care for APAP overdose [48]. However, the therapeutic effectiveness of NAC is inversely related to the time when NAC is administered after APAP overdose [49]. NAC also presents with some concerning side effects with up to 18% of patients receiving IV NAC reported anaphylactic reactions (rash, hypotension, wheezing, and shortness of breath) [50]. Oral administration, although effective, is not preferred due to low bioavailability and adverse events, such as nausea/vomiting and unpleasant taste [51]. Therefore, new therapies for APAP overdose are needed.

PG is a compendial pharmaceutical excipient. Products containing PG have been approved by FDA for clinical use [52]. PG has been shown to inhibit generation of NAPQI via inactivation of CYP2E1 activity [53,54]. Inhibition of CYP2E1 was accomplished by administering a pediatric preparation of APAP containing PG, which resulted in reduced CYP2E1-derived metabolites [55]. In another clinical trial, human subjects were randomized to receive ~50 mg/kg APAP daily in one arm and 50 mg/kg APAP in 5 mL of 99% PG daily in the other arm for 14 days. A >2-fold increase from baseline in ALT was considered as APAP responders. Although there was no difference in percentage of responders between the 2 arms, the mean percentage of CYP-derived metabolites was 5.8% (APAP) versus 4.3% (APAP+PG), *p* = 0.018. Among APAP responders, the mean percentage of CYP metabolites was 7.7% (APAP, *n* = 6 of 21) versus 4.6% (APAP+PG, *n* = 8 of 20), *p* = 0.050, whereas there was no difference among 2 arms of non-responders. The results indicate that PG inhibits CYP2E1-derived metabolites of APAP but does not affect hepato cellular injury at the given dose [54,55]. In the present study, PG significantly decreased APAP induced liver injury markers, LDH, ASP and ALT, in mice, showing that it protected liver from injury from APAP overdose. However, PG did not seem to prevent the activation of cell death pathways.

A recent publication demonstrates a detailed molecular mechanism by which 25HC3S functions as an endogenous epigenetic regulator [30]. The enzyme kinetic study demonstrated that 25HC3S specifically inhibited DNA methyltransferases, DNMT1, DNMT3a, and DNMT3b with IC_50_ at uM levels. In human hepatocytes, high glucose (HG) induces lipid accumulation by increasing promoter CpG methylation of key genes involved in the development of non-alcoholic fatty liver diseases (NAFLD) [18]. Using this model, whole genome bisulfate sequencing (WGBS) analysis demonstrated that 25HC3S converts HG-induced ^5m^CpG to CpG in the promoter regions of more than one thousand genes. Subsequently, 25HC3S increased expression of the demethylated genes, which are involved in the master signaling pathways, including MAPK-ERK, calcium-AMPK, and type II diabetes mellitus pathways. Messenger RNA array analysis showed that the upregulated genes encoded for key elements of cell survival [30]. The present study shows that 25HC3S protected organ function and reduced mortality via retaining mitochondrial polarization. The combination of 25HC3S, NAC and PG provided an optimal beneficial effect in APAP overdosed mice. Combined with the published results, a novel mechanism of the combination for effective therapy of APAP-induced ALF is proposed as shown in Figure 6. In this regard, PG inhibits the production of toxic metabolite, NAPQI; NAC increases the production of GSH that neutralizes the oxidants, including NAPQI; and 25HC3S stabilizes mitochondria, blocks cell death, and promotes cell survival by up-regulating important signaling pathways. 25HC3S has also been evaluated in vitro (cells) and other in vivo animal models, including NAFLD, NASH, hepatectomy, and acute multiple organ injury [25,29]. As an investigative drug, DUR-928 (25HC3S) has been in clinical trials in patients with NASH and acute alcohol-associated hepatitis [58,59,60,61]. Results from this study suggest that 25HC3S in PG and its appropriate combination have the potential in preventing or treating certain acute organ injuries.

## 5. Conclusions

25HC3S in PG and the combination with NAC in PG have the potential to serve as an effective biomedicine in the therapy of APAP-induced acute multiple organ injury.

## 6. Patents

Ren, S., Theeuwes, F., Brown, J.F. and Lin, W. Uses of oxygenated cholesterol sulfates (OCS). International Application: PCT/US2014/072128; U.S. Patent: 10,206,883, 10,786,517, 29 September 2020; Austria, 2014369916; Brazil, BR112016014611-5; Canada, 2932300; China, 201480070815.5; Europe, 14874617.5-1453, 19214560.5-1112; Israel, N/A; India, 201617019059; Japan, 2016-542662; South Korea, N/A; Mexico, MX/a/2016/008495; Taiwan, 103145013.

## Figures and Tables

**Figure 1 cells-10-03027-f001:**
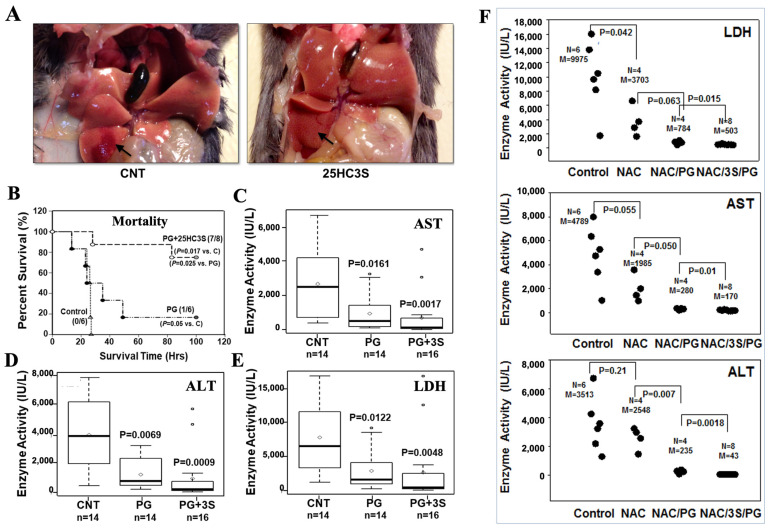
25HC3S treatment improves organ function and survival rates in APAP overdose mice. 12-week-old female C57BL/6J mice were administered either with control (*n* = 6), vehicle (*n* = 6) or 25HC3S (*n* = 8) 2 h before APAP (600 mg/kg) treatment. (**A**) The gross observation of liver after treated with 25HC3S in mice. The arrow indicates the site of APAP induced liver injury. (**B**) Mouse survival rates was observed and recorded up to 100 h. (**C**–**E**) The mice were injected with 350 mg/kg APAP and after half hour mice were intravenously treated with 10% glucose in sterile water for the control group, vehicle (20% propylene glycol (PG), 4% hydroxypropyl-β-cyclodextrin (HBC) in 10% glucose/water) for the PG group, and 25 mg/kg 25HC3S in vehicle for the 25HC3S group, respectively. After 24 h, serum activities of AST, ALT, and LDH were determined by a clinical laboratory. CNT: represents control mice with APAP injection only; PG: represents vehicle with PG treated control mice; 3S: 25HC3S treated mice. Solid bar shows the average value of each group. (**F**) Effects of PG, 25HC3S in PG, NAC, NAC in PG, and NAC+25HC3S in PG on injured liver function using the models as shown in (**C**–**E**). Each point represents an individual mouse and data are pooled from three independent experiments.

**Figure 2 cells-10-03027-f002:**
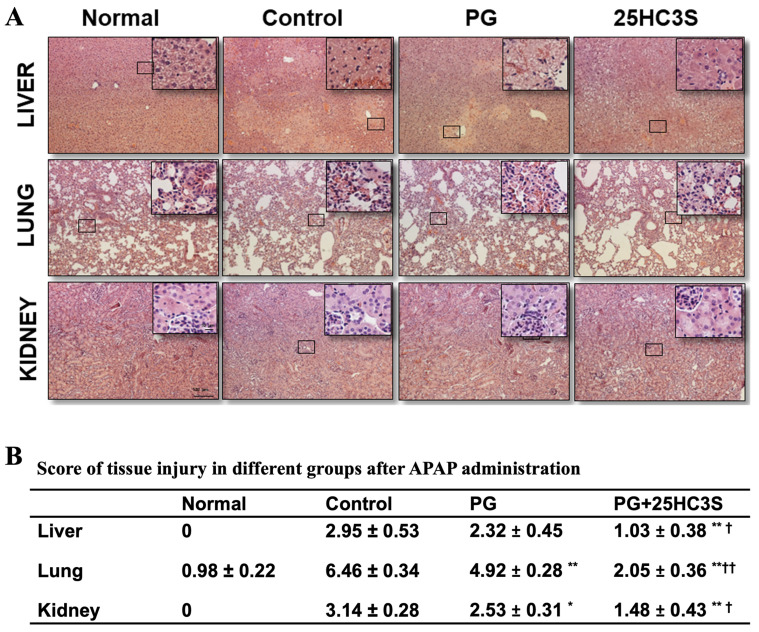
Morphological study of liver, lung, and kidney tissues. 12-week-old female C57BL/6J mice were administered either with control, vehicle or 25HC3S at 2 h before APAP (600 mg/kg) treatment (*n* = 4 for each group). The liver, lung, and kidney tissues were harvested at 24 h or at dying following the injection of APAP for morphological study. Tissues from age-matched mouse without any treatment were used as normal control. (**A**) The paraffin-embedded tissue sections were stained using H&E method and photographed for evaluation. Representative photos are shown at ×100 magnification (bar = 100 µm). Inserts are shown at ×400 magnification of the boxed areas (bar = 10 µm). Normal represents normal mice without any treatment (*n* = 4); Control, PG vehicle-treated mice; 25HC3S, 25HC3S-pretreated mice. (**B**) Ten images per sample were taken at ×400 magnification by light microscope and scored by two pathologists in a blinded manner. The severity of microscopic tissue injury was graded as indicated. Normal: normal mice without treatment; Control: mice with PG vehicle and APAP injection; 25HC3S: mice with 25HC3S and LPS injection (*n* = 4). The symbol * indicates *p* < 0.05 and ** indicates *p* < 0.01 versus Control group; † indicates *p* < 0.05 and †† indicates *p* < 0.01 versus PG group.

**Figure 3 cells-10-03027-f003:**
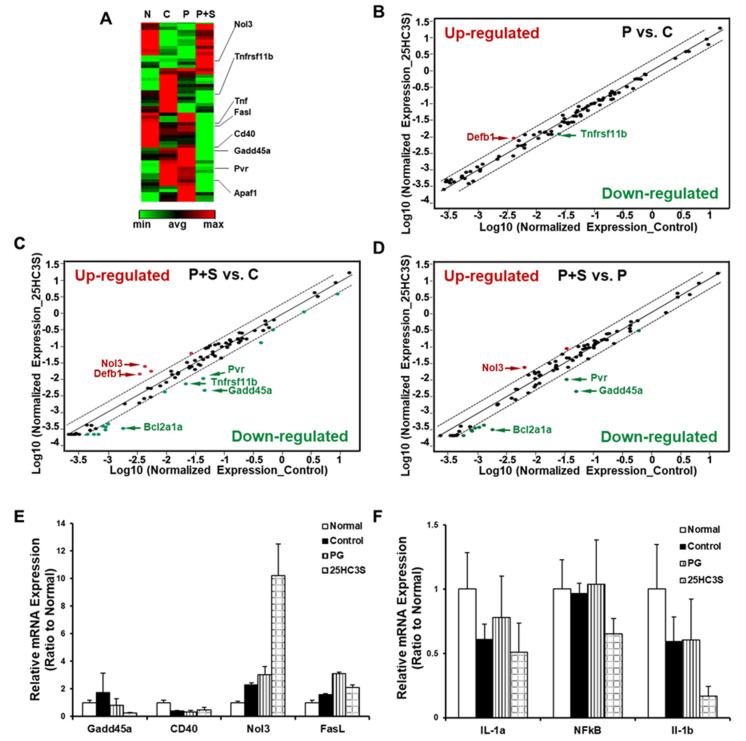
25HC3S treatment regulates expression of apoptosis-related genes in the liver tissues of APAP overdose mice. 12-week-old male C57BL/6J mice were intraperitoneally injected with 350 mg/kg APAP, half an hour later, mice were intravenously injected with 10% glucose in sterile water for the control group, vehicle (20% propylene glycol, 4% hydroxypropyl-β-cyclodextrin (HBC) in 10% glucose/water) for the PG group, and 25 mg/kg 25HC3S in vehicle for the 25HC3S group respectively. Mice without any treatment were used as a normal control. Each group contained four mice. Twenty-four hours following APAP injection, liver tissues were harvested, total mRNAs were extracted, four samples from each group were combined, and gene expressions were determined by the RT^2^ Profiler PCR Array assay. (**A**) Clustergram analysis of gene expression profiles: Lane N represents normal mice without any injection; Lane C, control mice with only APAP injection; Lane P, vehicle with PG pretreated; Lane S, 25HC3S-pretreated mice. (**B**) (PG vs. Control), (**C**) (25HC3S vs. Control), and (**D**) (25HC3S vs. PG) show results of scatter plot analysis: gene expressions with a greater than 2-fold change are highlighted. (**E**) qRT-PCR analysis to confirm the results of RT^2^ Profiler PCR Array assay. (**F**) qRT-PCR analysis to determine the expression of inflammation related genes.

**Figure 4 cells-10-03027-f004:**
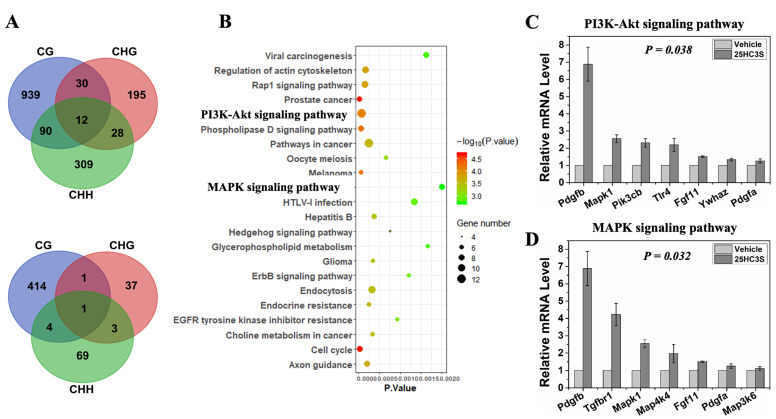
Effects of 25HC3S on DNA methylation in APAP induced liver injury mouse model by global methylation sequencing analysis. 12-week-old male C57BL/6J mice were intraperitoneally injected with 350 mg/kg APAP, half an hour later, mice were intravenously injected with 20% PG, 4% hydroxypropyl-β-cyclodextrin (HBC) in 10% glucose/water as the vehicle group, and 25 mg/kg 25HC3S in vehicle for the 25HC3S group. The mice were sacrificed at 24 h following the treatment. Total DNA were extracted from 25 mg of liver tissues, and 5.2 µg of the extracted DNA were used to create the whole genome bisulfite sequencing libraries. (**A**) Venn diagrams of hypomethylated DMR-associated genes (DMGs) in 25HC3S and vehicle libraries under CG, CHG, and CHH contexts of whole genome (Up) and promoter regions (Low). KOBAS software was used to test the statistical enrichment of DMR related genes in the Kyoto Encyclopedia of Genes and Genomes (KEGG) pathways. (**B**) High enrichment of hypomethylated DMRs in promoter regions in KEGG pathways. The detailed KEGG pathways as shown in Appendix A. (**C**) Represents the gene expression, mRNA levels of demethylated genes involved in PI3K-Akt signaling pathway; and (**D**) MAPK signaling pathway.

**Figure 5 cells-10-03027-f005:**
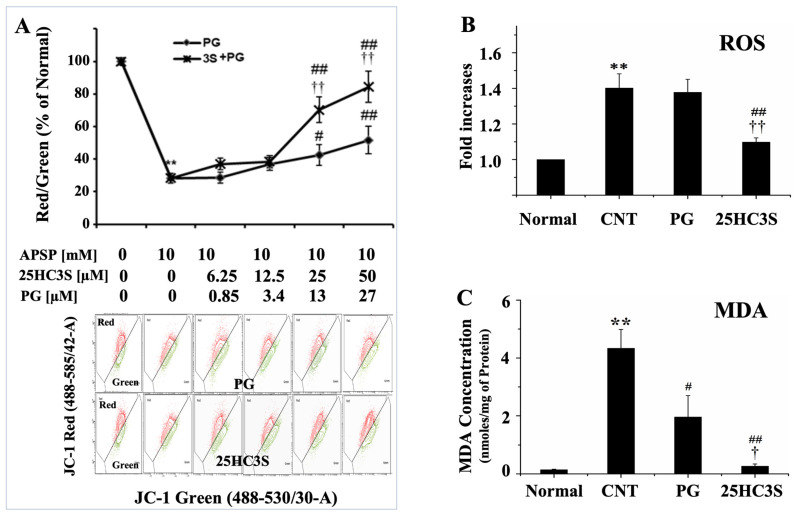
25HC3S restores the injured mitochondria membrane potential and decreases oxidant levels. (**A**) shows effects of 25HC3S on mitochondrial polarization, Huh-7 cells were seeded in 60 mm dishes at 6.5 × 10^5^/dish and cultured overnight before different dosages of 25HC3S (6.25, 12.5, 25, 50 µM in PG) were added, the relative amount PG were added as control (0.85, 3.4, 13, and 27 µM). Two hours later, cells were treated with 10 mM APAP for 24 h and harvested. Cells without any treatment were used as control. MMP was analyzed by JC-1 staining on flow cytometer using 488 nm excitation with 530/30 nm (Green) and 585/42 nm (Red) band-pass emission filters. MMP was indicated by red/green fluorescence ratio. (**B**) shows 25HC3S significantly decreases ROS levels in Huh-7 cells. Huh-7 cells were pretreated with the 50 µM 25HC3S and/or vehicle for 2 h before 10 mM APAP was added. Sixteen hours after APAP addition, the cells were harvested. H2DCFDA method was used to detect ROS levels by flow cytometry. Results were expressed as the relative changes compared to untreated cells. (**C**) shows MDA levels in liver tissues. Mice were challenged with 350 mg/kg APAP for 30 min and treated with 25 mg/kg 25HC3S for 24 h. The liver tissues were harvested and ~100 mg was homogenized. MDA were extracted and determined by the bicinchoninic acid assay kit purchased from Pierce (Rockford, IL, USA). The amount of lipid peroxidation, MDA, was expressed as nmol/mg protein. Data represent the mean ± SD for three independent experiments. The symbol ** indicates *p* < 0.01 versus Normal group; # indicates *p* < 0.05 and ## indicates *p* < 0.01 versus CNT group and †† indicates *p* < 0.01 versus PG group.

**Figure 6 cells-10-03027-f006:**
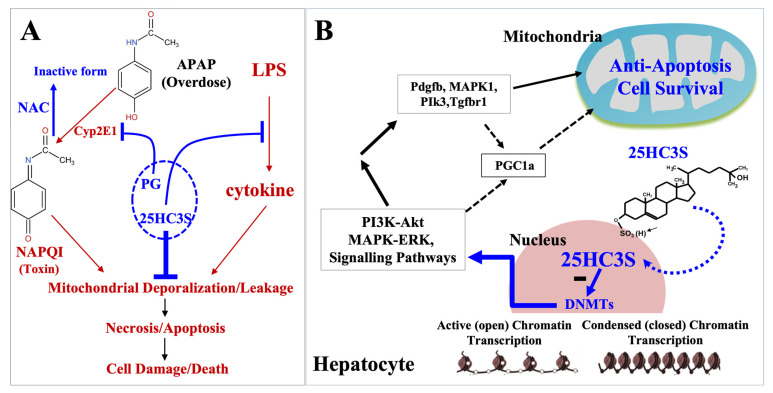
Schematic diagram depicting pharmaceutical mechanism. PG suppresses generation of NAPQI by inhibiting Cyp2E1 activity [55]. NAC is to promote hepatic glutathione (GSH) synthesis, which supports the detoxification of NAPQI and reduces protein binding [47,56,57]. The synergic effect of 25HC3S with NAC and PG on cell anti-apoptosis and cell survivals is shown in Panel (**A**). When 25HC3S enters to nuclear and potently inhibits DNMT-1, and 3a/3b, resulting in demethylation of ^5m^CpG in promoter regions of key genes involved in PI3K and MAPK signaling pathways. The increases in the expression of these key genes, *Fgf11*, *Pdgfa*, *Pdgfb*, *Map3k6*, *Map4k4*, *Ywhaz*, *Tgfbr*, *Tlr4*, *Pik3cb*, and *Mapk1* in the MAPK and PI3K-Akt pathways lead to mitochondrial stabilization and polarization, and subsequently decreases levels of oxidants, ROS and MDA, resulting in cell survival. Effects of combination of 25HC3S with N-acetyl cysteine (NAC) in propylene glycol (PG) on generation and neutralization of toxic APAP metabolite, NAPQI (**B**); and their role in maintenance of mitochondrial integrity and cell survival (anti-cell death). Solid bar represents that the data have been presented in the text, and dot bar, data have been published. Arrow represents activating, and bar, blocking.

**Table 1 cells-10-03027-t001:** Methylation of DMR in gene promoter regions of PI3K-AKt signaling pathway.

Gene Name	Gene Accession ID	DMR Location in Promoter Region			DMR (Methylation %)		
		Chromosome	Start	End	Vehicle	25HC3S	25HC3S-Vehicle
*Fgf11*	NM_010198	Chr11	69,802,413	69,802,474	55.0	22.6	−32.4
*Pik3cb*	NM_029094	Chr9	99,140,132	99,140,237	18.2	5.1	−13.1
*Gm26577*	AC_153914	Chr10	21,145,028	21,145,169	27.9	5.9	−22.0
*Mapk1*	NM_001038663	Chr16	16,983,558	16,983,663	11.5	4.8	−6.7
*Pdgfa*	NM_008808	Chr5	139,000,000	139,000,000	28.2	5.3	−22.9
*Pdgfb*	NM_011057	Chr15	80,013,952	80,014,060	37.0	6.9	−30.1
*Ppp2r5c*	NM_001081458	Chr12	110,000,000	110,000,000	47.0	6.6	−40.4
*Ccne2*	NM_009830	Chr4	11,191,626	11,191,706	25.6	7.7	−17.9
*Il3ra*	NM_008369	Chr14	14,346,685	14,346,808	32.2	17.1	−15.1
*Ywhaz*	NM_001253806	Chr15	36,793,096	36,793,150	43.7	7.3	−36.4
*Gsk3b*	NM_019827	Chr16	38,089,505	38,089,575	23.0	5.5	−17.5
*Tlr4*	NM_021297	Chr4	66,827,577	66,827,646	63.5	25.6	−37.9

**Table 2 cells-10-03027-t002:** Methylation of DMR in gene promoter regions of MAPK signaling pathway.

Gene Name	Gene Accession ID	DMR Location in Promoter Region			DMR (Methylation %)		
		Chromosome	Start	End	Vehicle	25HC3S	25HC3S-Vehicle
*Fgf11*	NM_010198	Chr11	69,802,413	6,9802,474	55.0	22.6	−32.4
*Map3k6*	NM_016693	Chr9	133,000,000	133,000,000	57.4	25.3	−32.1
*Mapk1*	NM_001038663	Chr10	16,983,558	16,983,663	11.5	4.8	−6.7
*Pdgfa*	NM_008808	Chr16	139,000,000	139,000,000	28.2	5.3	−22.9
*Pdgfb*	NM_011057	Chr5	80,013,952	80,014,060	37.0	6.9	−30.1
*Tgfbr1*	NM_009370	Chr15	47,353,529	47,353,605	45.8	8.2	−37.6
*Map4k4*	NM_001252200	Chr12	39,900,963	39,901,013	28.7	5.7	−23.0
*Ppm1a*	NM_008910	Chr4	72,761,171	72,761,247	27.2	5.6	−21.6

## Data Availability

All data described are contained within the article.

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
