# Peer review of "25-Hydroxycholesterol 3-Sulfate Recovers Acetaminophen Induced Acute Liver Injury via Stabilizing Mitochondria in Mouse Models"

_cells, 2021, doi:10.3390/cells10113027_

Round 1

Reviewer 1 Report

I would suggest to accept the submission in the current form. Some grammar errors appear in the revised (highlighted) statements.

Author Response

Response to Reviewer 1 Comments

Abstract:

Expand the abbreviations: MDA, ROS, PG.

These abbreviations have been expanded.

Methods:

Should be clearly distinguished that two modes of APAP and 25HC3S administration were applied (different mutual time relationship).

A good suggestion.  We have tried different doses and different time courses.  What we presented in the manuscript are optimized models.  We have now mentioned the methods.

Why for mortality study you applied first 25HC3S and then APAP; this model is quite different from all other experiments.

Yes, it is quite different.  In the mortality model, we used 600 mg/kg of APAP.  The mice died very soon after administration, and associated with a large variability in control groups.  Therefore, we cannot use the 600 mg/kg dose for liver function studies. We were able to control the times with use of 300 mg/ kg; allowing for  a good comparison.

Why you choose 30 min gap time between APAP and 25HC3S administration; it would be relevant also to see the results with a lag time of 4-6 hours (and that could better reflect clinical situation); at this time interval the observed effect of 25HC3S may be quite different.

Good question.  We did administration of 25HC3S at -2 hr, -1 hr, 0 hr, +30 min, +1 hr, and +2 hr before, on, and after injection APAP.  We found 25HC3S administration at 30 min after APAP injection gave the best results in terms of clinical significance.  Of course it is the earlier, the better.    We have mentioned the point in the results now.

Results:

As above. I would suggest to split Fig. 1 into two, i.e. treatment experiment, and mortality experiment (different timing of exposure for APAP and 25HC3S according to the methods section). Panel F: Which group results it represents?

Thanks. We would have liked to split the figure. As a function of the limitation of figure numbers by editorial policy, we combined the results.  A more descriptive legend in now included.

Discussion:

Other inn used for acetaminophen is paracetamol, should be mentioned somewhere in discussion or introduction.

We have now discussed paracetamol and cited a reference in the introduction.

Reviewer 2 Report

It is the opinion of this reviewer that authors do not make sufficient efforts to address concerns that were raised in the context of their initially submitted manuscript.

Specifically, authors do not make convincing attempts to discriminate the protective effects of the solvent propylene glycol from those of 25HC3S. As outlined in the primary evaluation, this creates a conceptual problem that, in the opinion of this reviewer, needs to be addressed before publication.

Moreover, authors claim synergism between 25HC3S and NAC/propylene glycol. As outlined in the primary evaluation, this proposed synergism is not sufficiently documented in Fig. 1F. The difference between NAC/propylene glycol and NAC/propylene glycol/25HC3S is based on small numbers (ALT: 235 versus 43) and functionally not relevant in light of a very strong protective effect by NAC/propylene glycol alone (ALT: 3513 (APAP) versus 235 (APAP + NAC/propylene glycol). The suggested synergism must therefore be investigated in the context of suboptimal NAC dosing.

Author Response

Response to Reviewer 2 Commonts

In the present manuscript Wang et al. set out to investigate effects of 25-hydroxycholesterol 3-sulfate (25HC3S) on the course of acetaminophen (APAP)-induced liver injury. This specific aim is outlined in the manuscript title and abstract.

Major points

1) To do so, authors use the vehicle propylene glycol (PG) in order to solubilize 25HC3S. Authors quote and confirm previous observation that PG (alone) mediates significant protection in murine APAP-induced acute liver injury (ALI) (see e.g. Hughes et al. Biochem. Pharmacol (1991) 42:710-13; and reference 51). Published data suggest that this protective property of PG is achieved by inhibition of APAP-metabolizing Cyp2e1.

It is the opinion of this reviewer that the experimental protocol used herein altogether creates a substantial conceptual problem because authors cannot exclude that 25HC3S’s effects depend on cooperative action with protective PG. This issue precludes publication of the manuscript in its current form.

Although this reviewer recognizes that this is an elaborate task, authors should identify a vehicle that, by itself, does not affect APAP intoxication. If that should actually be impossible, thorough dose-response-curves (of PG and 25HC3S) may reveal PG dosages that are pharmacologically inactive in the given context. Although this latter approach of dosage reductions (of PG and possibly also of 25HC3S) may not be the best option, it appears doable - if dose-dependent PG effects on APAP-induced ALI are adequately explained, shown, and discussed in an accordingly adapted manuscript.

In any case, the study needs farreaching conceptional adaptations.

Good point.  We understand the reviewer’s concern.  Decreases in generation of toxic products from APAP by PG has been widely investigated (we have added the reference as suggested as well).  However, the results from clinical trial did not show that PG therapy is promising as discussed.  As we showed in Fig. 1, PG did not affect mortality rates when it is induced by high doses of APAP (600 mg/kg), indicating PG alone cannot be an effective therapy. Interestingly, treatment with 25HC3S in PG significantly decreases mortality, indicating 25HC3S is able to block hepatocyte apoptosis and necrosis as shown in our previous publication (Ning, Y. et al “Cholesterol Metabolites Alleviate Injured Liver Function and Decrease Mortality in LPS-induced Mouse Model. Metabolism: Clinical and Experimental, 71, 83-93, 2017”.  Furthermore, we have recently published a detailed mechanism for how 25HC3S blocks hepatocyte apoptosis and induces cell proliferation “25HC3S is an endogenous ligand of DNMT in hepatocytes” in Journal of Lipid Research (2021)62, 100063, which is highlighted by ASBMB Today (American Society of Biochemistry and Molecular Biology), August 3, 2021.  We have added the detail mechanism in the discussion section in the current version.  In this manuscript, we have shown that 25HC3S recovers APAP via stabilizing mitochondrial integrity and retaining their polarization.  The results further provide evidence that 25HC3S has potent function of preventing cell death.  Combined with the published results (we have added the information in the discussion section), we made this conceptional mechanism (Fig. 6).                 

2) It is consensus in the field that mice should be starved before APAP administration. Authors must clarify why experiments were performed with un-starved animals.

It is true that the published mouse models used starved mice before APAP administration to increase APAP toxicity.  We believe that the fasting itself causes liver damage, which will make the models more complicated. Giving 10% glucose will eliminate the starving damage and focus on APAP toxicity.  We believe that this is a better model for the study of APAP toxicity, which is also more similar to clinical situation when treating patients with 10% glucose IV injection.  The results have shown that these models gave us more consistent data and proved this should be a better model for the study of APAP toxicity.   

3) p. 6/7:

‘The addition of 25HC3S to NAC+PG virtually restored LDH, AST, and ALT to the normal levels with P values of 0.015, 0.01, and 0.002, respectively (Figure 1F), indicating that the combination has potential as an optimal therapy of APAP induced acute liver injury.’

p.14:

‘The present results also provide strong evidence that the combination of 25HC3S+NAC in PG solution may have synergistic effects on the recovery of APAP induced acute organ injury as shown in Figure 6A, especially the liver injury.’

Both statements/conclusions are not based on data and incomprehensible for the reader. In fact, Figure 1F indicates no visible (biologically relevant) difference between the NAC/PG- and NAC/PG/25HC3S treatment-groups with regard to disease parameters such as LDH, AST, and ALT – which is due to an impressive protective/cooperative effect of PG on NAC-treatment!

PG and NAC do have cooperative effect, one is reducing agents and the other is inhibitor of CYP2E1.  However, as soon as toxic product, NAPQI, generated, NAC/PG has no effect on the mortality because none of them has the function of anti-apoptosis.  Thus, 25HC3S is necessary for the effective therapy as discussed in the discussion section. NAC/PG/25HC3S significantly decreases ALT much more than NAC/PG, 43 against 235 and all the other markers are statistically significant.  

Reviewer 3 Report

This work was addressed to investigate the protective mechanisms activated by 25‐Hydroxycholesterol 3‐Sulfate (25HC3S) on the adverse effects induced by the acetaminophen in the mouse liver. The investigation shows interesting findings; however, some issues need to be addressed before publication.

For in vitro experiments, authors used 10 mM of APAP; however, they did not explain/discuss whether that concentration is equivalent to the APAP overdose used in animals.

The inclusion of images from the lung and kidney takes the research out of context as the authors did not conduct additional experiments on the effect of 25HC3S on these organs. Furthermore, the authors also did not discuss anything about that phenomenon. So, my suggestion is that images from those organs should be excluded from the manuscript and instead, authors should improve the liver images in terms of size and quality.

NCBI Accession Numbers for genes in Table 1, Table 2, Table S1 must be included.

Although the authors endeavored to show their main findings in a graphical summary (Figure 6), they included some molecules that were not determined in liver samples such as LPS and NAPQI. The graphical summary should only show those molecules associated to the findings; so, the other should be either deleted or clearly indicated in figure legend that they were included based on previous reports.

Based on HUGO Gene Nomenclature Committee, gene and protein symbols must be indicated according to the accepted nomenclatures. So, throughout the manuscript including main text, figures, tables, and figures legends, gene and protein symbols must be properly indicated. As reference, authors should review HUGO website as well as the following article: PMID: 22836666.

Line 145: “This PCR array profiles the 84 key genes that play important roles in central mechanisms of cellular death” Something is missing in this sentence. Please, read it carefully and correct it.

Author Response

Response to Reviewer 3 Comments

  • For in vitro experiments, authors used 10 mM of APAP; however, they did not explain/discuss whether that concentration is equivalent to the APAP overdose used in animals.

Human hepatocytes are more sensitive to APAP than mice.  The concentration of 10 mM APAP has been widely used in vitro human hepatocyte models.  A reference has been cited in Methods.

  1. The inclusion of images from the lung and kidney takes the research out of context as the authors did not conduct additional experiments on the effect of 25HC3S on these organs. Furthermore, the authors also did not discuss anything about that phenomenon. So, my suggestion is that images from those organs should be excluded from the manuscript and instead, authors should improve the liver images in terms of size and quality.

Good points. We have consulted with pathologists and physicians.  They prefer to see the images as more direct evidence for improvements. As suggested, we have discussed the recovery of these multiple organ injury in the Discussion Section.

  1. NCBI Accession Numbers for genes in Table 1, Table 2, Table S1 must be included.

All NCBI Accession Numbers have been added.

  1. Although the authors endeavored to show their main findings in a graphical summary (Figure 6), they included some molecules that were not determined in liver samples such as LPS and NAPQI. The graphical summary should only show those molecules associated to the findings; so, the other should be either deleted or clearly indicated in figure legend that they were included based on previous reports.

Good suggestion.  We have indicated where the molecules come from and cited all the reference to the molecules in the legend.  Thanks

  1. Based on HUGO Gene Nomenclature Committee, gene and protein symbols must be indicated according to the accepted nomenclatures. So, throughout the manuscript including main text, figures, tables, and figures legends, gene and protein symbols must be properly indicated. As reference, authors should review HUGO website as well as the following article: PMID: 22836666.

We did as suggested.

  1. Line 145: “This PCR array profiles the 84 key genes that play important roles in central mechanisms of cellular death” Something is missing in this sentence. Please, read it carefully and correct it.

We have carefully read and corrected as "this PCR array is to detect the 84 key gene expression (mRNA levels), which are involved in central mechanisms of cellular death including apoptosis, autophagy, and necrosis."  Thanks.

Round 2

Reviewer 3 Report

The authors have properly accomplished all comments and suggestions. So, the revised version of the manuscript ID cells-1422419 is now acceptable to be considered for publication by Cells journal.

This manuscript is a resubmission of an earlier submission. The following is a list of the peer review reports and author responses from that submission.

Round 1

Reviewer 1 Report

I have the following comments and suggestions to the submission:

Abstract:

Expand the abbreviations: MDA, ROS, PG.

Methods:

Should be clearly distinguished that two modes of APAP and 25HC3S administration were applied (different mutual time relationship).

Why for mortality study you applied first 25HC3S and then APAP; this model is quite different from all other experiments.

Why you choose 30 min gap time between APAP and 25HC3S administration; it would be relevant also to see the results with a lag time of 4-6 hours (and that could better reflect clinical situation); at this time interval the observed effect of 25HC3S may be quite different.

Results:

As above. I would suggest to split fig. 1 into two, i.e. treatment experiment, and mortality experiment (different timing of exposure for APAP and 25HC3S according to the methods section). Panel F: Which group results it represents?

Discussion:

Other inn used for acetaminophen is paracetamol, should be mentioned somewhere in discussion or introduction.

Reviewer 2 Report

In the present manuscript Wang et al. set out to investigate effects of 25-hydroxycholesterol 3-sulfate (25HC3S) on the course of acetaminophen (APAP)-induced liver injury. This specific aim is outlined in the manuscript title and abstract.

Major points

1) To do so, authors use the vehicle propylene glycol (PG) in order to solubilize 25HC3S. Authors quote and confirm previous observation that PG (alone) mediates significant protection in murine APAP-induced acute liver injury (ALI) (see e.g. Hughes et al. Biochem. Pharmacol (1991) 42:710-13; and reference 51). Published data suggest that this protective property of PG is achieved by inhibition of APAP-metabolizing Cyp2e1.

It is the opinion of this reviewer that the experimental protocol used herein altogether creates a substantial conceptual problem because authors cannot exclude that 25HC3S’s effects depend on cooperative action with protective PG. This issue precludes publication of the manuscript in its current form.

Although this reviewer recognizes that this is an elaborate task, authors should identify a vehicle that, by itself, does not affect APAP intoxication. If that should actually be impossible, thorough dose-response-curves (of PG and 25HC3S) may reveal PG dosages that are pharmacologically inactive in the given context. Although this latter approach of dosage reductions (of PG and possibly also of 25HC3S) may not be the best option, it appears doable - if dose-dependent PG effects on APAP-induced ALI are adequately explained, shown, and discussed in an accordingly adapted manuscript.

In any case, the study needs farreaching conceptional adaptations.

2) It is consensus in the field that mice should be starved before APAP administration. Authors must clarify why experiments were performed with un-starved animals.

3) p. 6/7:

‘The addition of 25HC3S to NAC+PG virtually restored LDH, AST, and ALT to the normal levels with P values of 0.015, 0.01, and 0.002, respectively (Figure 1F), indicating that the combination has potential as an opti-mal therapy of APAP induced acute liver injury.’

p.14:

‘The present results also provide strong evidence that the combination of 25HC3S+NAC in PG solution may have synergistic effects on the recovery of APAP induced acute organ injury as shown in Figure 6A, especially the liver injury.’

Both statements/conclusions are not based on data and incomprehensible for the reader. In fact, Figure 1F indicates no visible (biologically relevant) difference between the NAC/PG- and NAC/PG/25HC3S treatment-groups with regard to disease parameters such as LDH, AST, and ALT – which is due to an impressive protective/cooperative effect of PG on NAC-treatment!